# SCAN: Structure Correcting Adversarial Network for Organ Segmentation in Chest X-rays

**Wei Dai, Nanqing Dong, Zeya Wang, Xiaodan Liang,**
**Hao Zhang, Eric P. Xing**
Petuum Inc.
`wei.dai,nanqing.dong,zeya.wang,xiaodan.liang,`
`hao.zhang,eric.xing@petuum.com`

## Abstract

Chest X-ray (CXR) is one of the most commonly prescribed medical imaging procedures, often with over 2–10x more scans than other imaging modalities. These voluminous CXR scans place significant workloads on radiologists and medical practitioners. Organ segmentation is a key step towards effective computer-aided detection on CXR. In this work, we propose Structure Correcting Adversarial Network (SCAN) to segment lung fields and the heart in CXR images. SCAN incorporates a critic network to impose on the convolutional segmentation network the structural regularities inherent in human physiology. Specifically, the critic network learns the higher order structures in the masks in order to discriminate between the ground truth organ annotations from the masks synthesized by the segmentation network. Through an adversarial process, the critic network guides the segmentation network to achieve more realistic segmentation that mimics the ground truth. Extensive evaluation shows that our method produces highly accurate and realistic segmentation. Using only very limited training data available, our model reaches human-level performance without relying on any pre-trained model. Our method surpasses the current state-of-the-art and generalizes well to CXR images from different patient populations and disease profiles.

## 1  Introduction

Chest X-ray (CXR) is one of the most common medical imaging procedures. Due to CXR's low cost and low dose of radiation, hundreds to thousands of CXRs are generated in a typical hospital daily, which create significant diagnostic workloads. In 2015/16 year over 22.5 million X-ray images were requested in UK's public medical sector, constituting over 55% of the total number of medical images and dominating all other imaging modalities such as computed tomography (CT) scan (4.5M) and MRI (3.1M) [1]. Among X-ray images, 8 million are Chest X-rays, which translate to thousands of CXR readings per radiologist per year. The shortage of radiologists is well documented across the world [2, 3]. It is therefore of paramount importance to develop computer-aided detection methods for CXRs to support clinical practitioners.

An important step in computer-aided detection on CXR images is organ segmentation. The segmentation of the lung fields and the heart provides rich structural information about shape irregularities and size measurements [4] that can be used to directly assess certain serious clinical conditions, such as cardiomegaly (enlargement of the heart), pneumothorax (lung collapse), pleural effusion, and emphysema. Furthermore, explicit lung region masks can also mask out non-lung regions to minimize the effect of imaging artifacts in computer-aided detection, which is important for the clinical use [5].

One major challenge in CXR segmentation is to incorporate the implicit medical knowledge involved in contour determination. For example, the heart and the lung contours should always be adjacent

1st Conference on Medical Imaging with Deep Learning (MIDL 2018), Amsterdam, The Netherlands.

Figure 1: Two example chest X-ray (CXR) images from two dataset: JSRT (top) and Montgomery (bottom). The left and right columns show the original CXR images and the lung field annotations by radiologists. JSRT (top) additionally has the heart annotation. Note that contrast can vary significantly between the dataset, and pathological lung profiles such as the bottom patient pose a significant challenge to the segmentation problem.

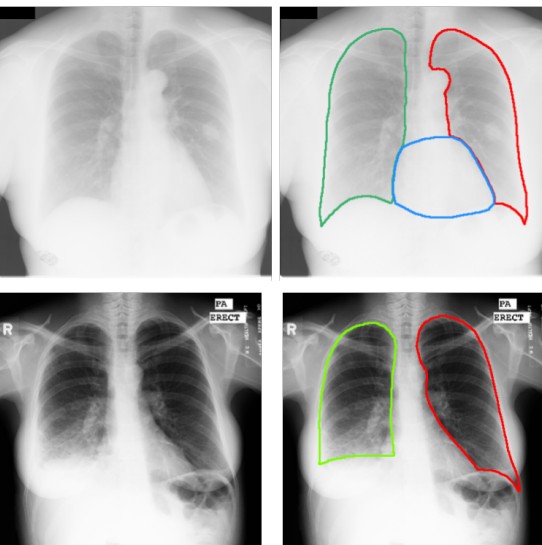

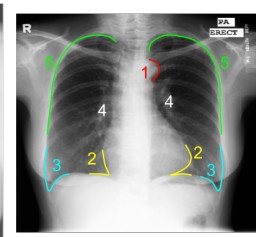

Figure 2: Important contour landmarks around lung fields: aortic arch (1) is excluded from lung fields; costophrenic angles (3) and cardiodiaphragmatic angles (2) should be visible in healthy patients. Hila and other vascular structures (4) are part of the lung fields. The rib cage contour (5) should be clear in healthy lungs.

to each other due to definition of the lung boundaries (Section 2). Moreover, when medical experts annotate the lung fields, they look for certain consistent structures surrounding the lung fields (Fig. 2). Such prior knowledge helps resolve ambiguous boundaries caused by pathological conditions or poor imaging quality, as can be seen in Fig. 1. Therefore, a successful segmentation model must effectively leverage global structural information to resolve the local details.

Unfortunately, unlike natural images, there are very limited CXR data because of sensitive privacy issues. Even fewer training data have pixel-level annotations, due to the expensive label acquisition involving medical professionals. Furthermore, CXRs exhibit substantial variations across different patient populations, pathological conditions, as well as imaging technology and operation. Finally, CXR images are gray-scale and are drastically different from natural images, which may limit the transferability of existing models. Existing approaches to CXR organ segmentation generally rely on hand-crafted features that can be brittle when applied to different patient populations, disease profiles, or image quality. Furthermore, these methods do not explicitly balance local information with global structure in a principled way, which is critical to achieving realistic segmentation outcomes suitable for diagnostic tasks.

In this work, we propose to use the Structure Correcting Adversarial Network (SCAN) framework that incorporates a critic network to guide the convolutional segmentation network to achieve accurate and realistic organ segmentation in chest X-rays. By employing a convolutional network approach to organ segmentation, we side-step the problems faced by existing approaches based on ad hoc feature extraction. Our convolutional segmentation model alone can achieve performance competitive with existing methods. However, the segmentation model alone cannot capture sufficient global structures to produce natural contours due to the limited training data. To impose regularization based on the physiological structures, we introduce a critic network which learns the higher order structures in the masks in order to discriminate between the ground truth organ annotations from the masks synthesized by the segmentation network. Through an adversarial training process, the critic network effectively transfers this learned global information back to the segmentation network to achieve realistic segmentation outcomes that mimic the ground truth.

Without using any pre-trained models, SCAN produces highly realistic and accurate segmentation even when trained on a very small dataset. With the global structural information, our segmentation model is able to resolve difficult boundaries that require a strong prior knowledge. SCAN improves the state-of-the-art lung segmentation methods and achieves performance competitive with human experts. We further show that SCAN is more robust than existing methods when applied to different patient populations. To our knowledge, this is the first successful application of convolutional neural networks (CNN) to CXR image segmentation, and our CNN-based method can be readily integrated with neural network solutions in computer-aided detection.

## 2 Structure Correcting Adversarial Network

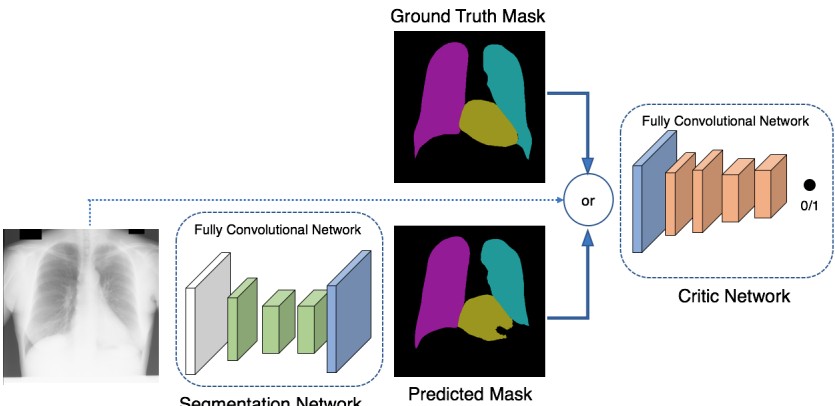

Figure 3: Overview of the proposed SCAN framework that jointly trains a segmentation network and a critic network with an adversarial mechanism. The segmentation network produces a per-pixel class prediction. The critic takes either the ground truth label or the prediction by the segmentation network, optionally with the CXR image, and output the probability estimate of whether the input is the ground truth (with training target 1) or the segmentation network prediction (with training target 0).

**Problem Definition.** We address the problem of segmenting the left lung field, the right lung field, and the heart on chest X-rays (CXRs) in the posteroanterior (PA) view, in which the radiation passes through the patient from the back to the front. Because CXR is a 2D projection of 3D structures, organs overlap significantly and one has to be careful in defining the lung fields. We adopt the definition from [6]: lung fields consist of all the pixels for which radiation passes through the lung but not through the following structures: the heart, the mediastinum (the opaque region between the two lungs), below the diaphragm, the aorta, and, if visible, the superior vena cava (Fig. 2). The heart boundary is generally visible on two sides, while the top and bottom borders of the heart have to be inferred due to occlusion by the mediastinum. This definition captures the common notion of lung fields and the heart, and include regions pertinent to CXR reading in the clinical settings (see Fig. 1).

Adversarial training was first proposed in Generative Adversarial Network (GAN) [7] in the context of generative modeling. The GAN framework consists of a generator network and a critic network that engage in an adversarial two-player game, in which the generator aims to learn the data distribution and the critic estimates the probability that a sample comes from the training data instead of synthesized by the generator. The generator's objective is to maximize the probability that the critic makes a mistake, while the critic is optimized to minimize the chance of mistake. It has been demonstrated that the generator produces samples (e.g., images) that are highly realistic [8].

A key insight in this adversarial process is that the critic, which itself can be a complex neural network, can learn to exploit higher order inconsistencies in the samples synthesized by the generator. Through the interplay of the generator and the critic, the critic can guide the generator to produce samples more consistent with higher order structures in the training samples, resulting in a more "realistic" data generation process.

The higher order consistency enforced by the critic is particularly desirable for CXR segmentation. Human anatomy, though exhibiting substantial variations across individuals, generally maintains a stable relationship between physiological structures (Fig. 2). CXRs also pose consistent views of these structures thanks to the standardized imaging procedures. We can, therefore, expect the critic to learn these higher order structures and guide the segmentation network to generate masks more consistent with the learned global structures.

We propose to use adversarial training for segmenting CXR images. Fig. 3 shows the overall SCAN framework in incorporating the adversarial process into the semantic segmentation. The framework consists of a segmentation network and a critic network that are jointly trained. The segmentation network makes pixel-level predictions of the target classes, playing the role of the generator in GAN but conditioned on an input image. On the other hand, the critic network takes the segmentation masks as input and outputs the probability that the input mask is the ground truth annotations instead

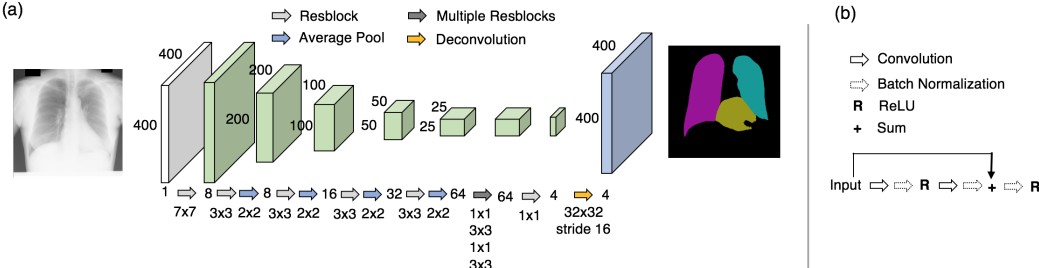

Figure 4: The segmentation network architecture. (a) Fully convolutional network for dense prediction. The feature map resolutions (e.g., 400×400) are denoted only for layers with resolutions different from the previous layer. The arrows denote the forward pass, and the integer sequence (1, 8, 16,...) is the number of feature maps. $k \times k$ below Resblock (residual block), average pool, and deconvolution arrows indicate the receptive field sizes. The dark gray arrow denotes 5 resblocks. All convolutional layers have stride $1 \times 1$, while all average pooling layers have stride $2 \times 2$. The output is the class distribution for 4 classes (3 foregrounds + 1 background). The total number of parameters is 271k, $\sim$ 500x smaller than the VGG-based down-sampling path in [9]. (b) The residual block architecture is based on [10]. The residual block maintains the same number of feature maps and spatial resolution which we omit here. Best viewed in color.

of the prediction by the segmentation network. The network can be trained jointly through a minimax scheme that alternates between optimizing the segmentation network and the critic network.

## 2.1 Training Objectives

Let $S$, $D$ be the segmentation network and the critic network, respectively. The data consist of the input images $\boldsymbol{x}_i$ and the associated mask labels $\boldsymbol{y}_i$, where $\boldsymbol{x}_i$ is of shape $[H, W, 1]$ for a single-channel gray-scale image with height $H$ and width $W$, and $\boldsymbol{y}_i$ is of shape $[H, W, C]$ where $C$ is the number of classes including the background. Note that for each pixel location $(j, k)$, $y_i^{jkc} = 1$ for the labeled class channel $c$ while the rest of the channels are zero ($y_i^{jkc'} = 0$ for $c' \neq c$). We use $S(\boldsymbol{x}) \in [0, 1]^{[H, W, C]}$ to denote the class probabilities predicted by $S$ at each pixel location such that the class probabilities normalize to 1 at each pixel. Let $D(\boldsymbol{x}_i, \boldsymbol{y})$ be the scalar probability estimate of $\boldsymbol{y}$ coming from the training data (ground truth) $\boldsymbol{y}_i$ instead of the predicted mask $S(\boldsymbol{x}_i)$. We define the optimization problem as

$$\min_S \max_D \left\{ J(S, D) := \sum_{i=1}^{N} J_s(S(\boldsymbol{x}_i), \boldsymbol{y}_i) - \lambda \Big[ J_d(D(\boldsymbol{x}_i, \boldsymbol{y}_i), 1) + J_d(D(\boldsymbol{x}_i, S(\boldsymbol{x}_i)), 0) \Big] \right\} \quad (1)$$

, where $J_s(\hat{\boldsymbol{y}}, \boldsymbol{y}) := \frac{1}{HW} \sum_{j,k} \sum_{c=1}^{C} -y^{jkc} \ln \hat{y}^{jkc}$ is the multi-class cross-entropy loss for predicted mask $\hat{\boldsymbol{y}}$ averaged over all pixels. $J_d(\hat{t}, t) := -\{t \ln \hat{t} + (1 - t) \ln(1 - \hat{t})\}$ is the binary logistic loss for the critic's prediction. $\lambda$ is a tuning parameter balancing pixel-wise loss and the adversarial loss. We can solve Eq. (1) by alternating between optimizing $S$ and optimizing $D$ with corresponding loss function.

**Training the Critic:** Since the first term in Eq. (1) does not depend on $D$, we can train our critic network by *minimizing* $\sum_{i=1}^{N} J_d(D(\boldsymbol{x}_i, \boldsymbol{y}_i), 1) + J_d(D(\boldsymbol{x}_i, S(\boldsymbol{x}_i)), 0)$ with respect to $D$; $S$ is fixed.

**Training the Segmentation Network:** Given a fixed $D$, we train the segmentation network by minimizing $\sum_{i=1}^{N} J_s(S(\boldsymbol{x}_i), \boldsymbol{y}_i) + \lambda J_d(D(\boldsymbol{x}_i, S(\boldsymbol{x}_i)), 1)$ with respect to $S$. Note that we use $J_d(D(\boldsymbol{x}_i, S(\boldsymbol{x}_i)), 1)$ in place of $-J_d(D(\boldsymbol{x}_i, S(\boldsymbol{x}_i)), 0)$, following the recommendation in [7]. This is valid as they share the same set of critical points. The reason for this substitution is that $J_d(D(\boldsymbol{x}_i, S(\boldsymbol{x}_i)), 0)$ leads to weaker gradient signals when $D$ makes accurate predictions, such as during the early stage of training.

## 2.2 Segmentation Network

Our segmentation network is a fully convolutional network (FCN), which is also the core component in many state-of-the-art semantic segmentation models [9, 11, 12]. The success of FCN can be largely attributed to convolutional neural network's excellent ability to extract high level representations suitable for dense classification. FCN consists of two modules: the down-sampling path and the up-sampling path. The down-sampling path consists of convolutional layers and max or average pooling

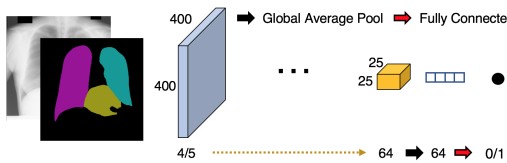

Figure 5: The critic network architecture. Our critic FCN mirrors the segmentation network (Fig. 4). The input to the critic network has 4 channels (one for each object masks). The subsequent layers are the same as the segmentation network (Fig. 4) up to the yellow box with 64 channel, which corresponds to the last green box in Fig. 4 with 64 channels. The training target is 0 for synthetic masks; 1 otherwise.

layers, with architecture similar to those used in image classification [13]. The down-sampling path can extract the high level semantic information, usually at a lower spatial resolution. The up-sampling path consists of convolutional and deconvolutional layers (also called transposed convolution) to predict scores for each class at the pixel level using the output of the down-sampling path.

Our FCN is highly parsimonious compared with existing FCNs for several reasons: (1) Small dataset: our training dataset of 247 CXR images is orders of magnitude smaller than the dataset in the natural image domains. (2) Object variety: in our task we focus on segmenting three classes (the left lung, the right lung, and the heart), which is a smaller classification space compared with dataset such as PASCAL VOC which has 20 classes of objects. (3) CXR is gray-scale with consistent viewpoint, which can be captured by fewer feature maps and thus fewer parameters.

Figure 4 details our FCN architecture. We start with just 8 feature maps in the first layers, compared with 64 feature maps in the first layer of VGG [13]. To obtain sufficient model capacity, we instead go deep with 20 convolutional layers. We also interleave $1 \times 1$ convolution with $3 \times 3$ in the final layers to emulate the bottleneck design [14]. All in all the segmentation network contains 271k parameters, 500x smaller than VGG-based FCN [9]. We employ residual blocks [14] (Fig. 4(b)) to aid optimization. The parsimonious network construction allows us to optimize it efficiently without relying on any existing trained model, which is not readily available for the medical domain.

### 2.3 Critic Network

Our critic network mirrors the construction of the segmentation network and is also a fully convolutional network. Fig. 5 shows the architecture, omitting the intermediate layers that are identical to the segmentation network. This way the critic network contains similar model capacity as the segmentation network and the later layers maintain a large field of view, which is important to capture the large lung fields in the CXR images. We can optionally include the original CXR image as input to the critic as an additional channel. We found that including the original CXR image does not improve performance, and for simplicity, we feed only the mask prediction to the critic network. Overall our critic network has 258k parameters, comparable to the segmentation network.

## 3 Experiments

We perform an extensive evaluation of the proposed SCAN framework and demonstrate that our approach produces highly accurate and realistic segmentation of CXR images.

### 3.1 Dataset and Protocols

We use two publicly available datasets to evaluate our proposed SCAN framework[1]. The datasets come from two different countries with different lung diseases, representing diverse CXR samples.

**JSRT.** The JSRT dataset was released by Japanese Society of Radiological Technology (JSRT) [15] and the lung fields and the heart masks labels were provided by [6] (Fig. 1). The dataset contains 247 CXRs, among which 154 have lung nodules and 93 have no lung nodule. All images have resolution $2048 \times 2048$ in gray-scale with a color depth of 12 bits.

**Montgomery.** The Montgomery dataset, collected in Montgomery County, Maryland, USA, consists of 138 CXRs, including 80 normal patients and 58 patients with manifested tuberculosis (TB) [16]. The CXR images are 12-bit gray-scale images of dimension $4020 \times 4892$ or $4892 \times 4020$. Only the lung masks annotations are available (Fig. 1).

---

[1]To our knowledge, these are the only two publicly available datasets with lung field annotations.

We scale all images to $400 \times 400$ pixels, which retains visual details for vascular structures in the lung fields and the boundaries. We present the details of data processing in Supplementary Material. We use the following two metrics for evaluation: **Intersection-over-Union (IoU)** is the agreement between the ground truth and the estimated segmentation mask. Formally, let $P$ be the set of pixels in the predicted segmentation mask for a class and $G$ the set of pixels in the ground truth mask for the same class. IoU is defined as $\frac{|P \cap G|}{|P \cup G|}$. **Dice Coefficient** is a popular metric for segmentation in the medical domain. Using the notations defined above, Dice coefficient can be calculated as $\frac{2|P \cap G|}{|P|+|G|}$.

## 3.2 Experiment Design and Results

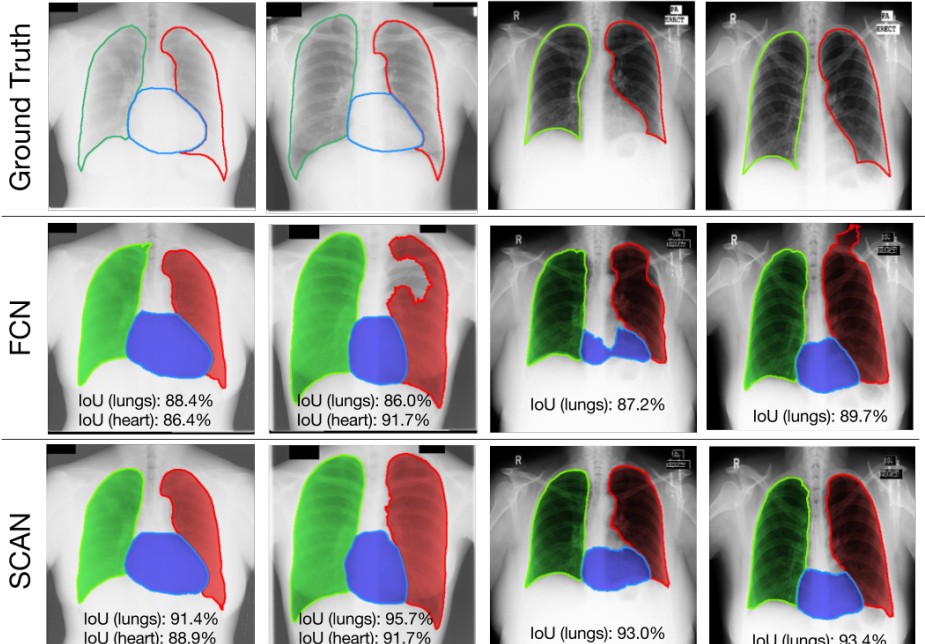

Figure 6: Visualization of segmentation results on 4 patients, one per column. From top to bottom: ground truth, FCN prediction without adversarial training, FCN prediction with adversarial training. The contours of the predicted masks are added for visual clarity. Each column is a patient. The left two columns are patients from the JSRT evaluation set with models trained on JSRT development set. The right two columns are from the Montgomery dataset using a model trained on the full JSRT dataset but not Montgomery, which is a much more challenging scenario. Note that only the two patients from JSRT dataset (left two columns) have heart annotations for evaluation of heart area IoU.

We randomly split the JSRT dataset into the development set (209 images) and the evaluation set (38 images). We tune our architecture and hyperparameter $\lambda$ (Eq. (1)) using a validation set within the development set. We use FCN to denote the segmentation network only architecture, and SCAN to denote the full framework with the critic.

**Quantitative Comparison.** We find that the performance of adversarial training is robust across a range of $\lambda$: the IoU for both lungs with $\lambda = 0.1, 0.01, 0.001$ is $94.4\% \pm 0.4\%$, $94.5\% \pm 0.4\%$, and $94.7\% \pm 0.4\%$, respectively. We use $\lambda = 0.01$ in SCAN for the rest of experiments.

We compare FCN with SCAN on the JSRT dataset. Table 1 shows the IoU and Dice scores. We observe that the adversarial training significantly improves the performance. In particular, IoU for the two lungs improves from 92.9% to 94.7%.

Table 2 compares our approach to several existing methods on the JSRT dataset, as well as human performance. Our model surpasses the current state-of-the-art method based on registration-based model [16] by a significant margin. Additionally, we compare with other standard CNN approaches for semantic segmentation: DeepLabV2 with ResNet101 [17] and U-Net [18] and demonstrate the advantage of our parsimonious architecture and adversarial training. Importantly, our method is competitive with the human performance for both lung fields and the heart.

|     |            | FCN              | SCAN             |
| --- | ---------- | ---------------- | ---------------- |
| IoU | Left Lung  | $91.3\% \pm 0.9\%$ | $\mathbf{93.8}\% \pm 0.8\%$ |
|     | Right Lung | $94.2\% \pm 0.2\%$ | $\mathbf{95.5}\% \pm 0.2\%$ |
|     | Both Lungs | $92.9\% \pm 0.5\%$ | $\mathbf{94.7}\% \pm 0.4\%$ |
|     | Heart      | $86.5\% \pm 0.9\%$ | $86.6\% \pm 1.2\%$ |
| Dice | Left Lung  | $95.4\% \pm 0.5\%$ | $\mathbf{96.8}\% \pm 0.5\%$ |
|     | Right Lungs | $97.0\% \pm 0.1\%$ | $\mathbf{97.7}\% \pm 0.1\%$ |
|     | Both Lungs | $96.3\% \pm 0.3\%$ | $\mathbf{97.3}\% \pm 0.2\%$ |
|     | Heart      | $92.7\% \pm 0.6\%$ | $92.7\% \pm 0.2\%$ |

Table 1: IoU and Dice scores on JSRT evaluation set for left lung (on the right side of the PA view CXR), right lung (on the left side of the image), both lungs, and the heart. The model is trained on the JSRT development set. $\pm$ represents one standard deviation estimated from bootstrap.

|                        | IoU (Lungs)       | IoU (Heart)        |
| ---------------------- | ----------------- | ------------------ |
| Human Observer [6]     | $\mathbf{94.6}\% \pm 1.8\%$ | $\mathbf{87.8}\% \pm 5.4\%$ |
| **Ours (SCAN)**        | $\mathbf{94.7}\% \pm 0.4\%$ | $86.6\% \pm 1.2\%$ |
| Registration-based [16] | $92.5\% \pm 0.4\%$ | –                  |
| DeepLabV2 101 [17]     | $85.7\% \pm 0.9\%$ | –                  |
| U-net [18]             | $84.4\% \pm 1.3\%$ | –                  |
| ShRAC [19]             | $90.7\% \pm 3.3\%$ | –                  |
| ASM [6]                | $90.3\% \pm 5.7\%$ | $79.3\% \pm 11.9\%$ |
| AAM [6]                | $84.7\% \pm 9.5\%$ | $77.5\% \pm 13.5\%$ |
| Mean Shape [6]         | $71.3\% \pm 7.5\%$ | $64.3\% \pm 14.7\%$ |

Table 2: Comparison with existing single-model approaches to lung field segmentation on JSRT dataset. Note that [6, 19] use different data splits than our evaluation.

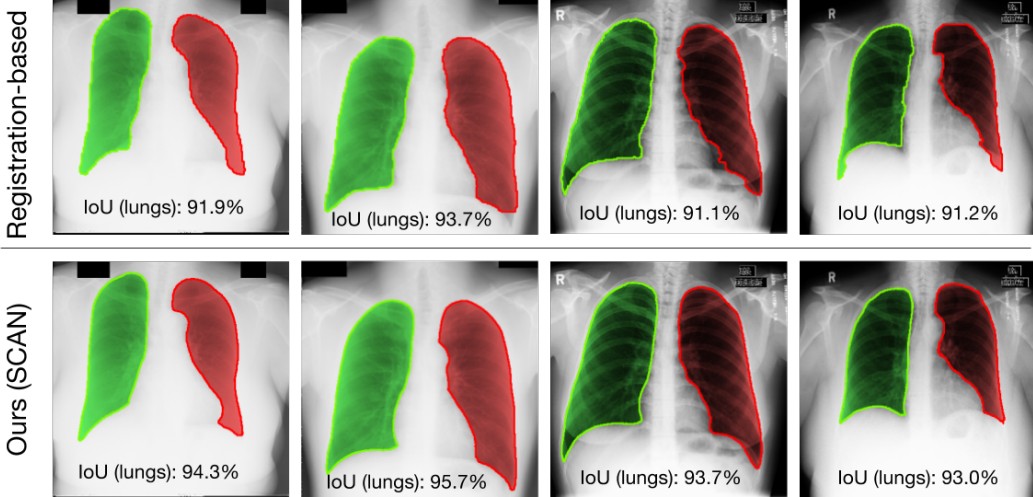

Figure 7: Comparison with the current state-of-the-art using a registration-based method [16]. SCAN produces sharp contours at the costophrenic angles for the left two columns (from the JSRT evaluation set). Furthermore, our model generalizes well to a different patient populations and imaging setup, as shown in the Montgomery CXR in the right two columns. [16] struggles due to the mismatch between train and test patient lung profiles (JSRT and Montgomery dataset, respective).

For clinical deployment, it is important for the segmentation model to generalize to a different population with different patient population and image qualities, such as when deployed in another country or a specialty hospital with very different disease distributions. In our next experiment, we therefore train our model on the full JSRT dataset, which is collected in Japan from a population with lung nodules, and test the model on the full Montgomery dataset, which is collected in the U.S. from patients potentially with TB. The two datasets present very different contrast (Fig. 1) and diseases. Table 3 shows that FCN alone does not generalize well to a new dataset, as IoU for both lungs degrades to $87.1\%$. However, SCAN substantially improves the performance, surpassing [16].

We further investigate the scenario when training on the two development sets from JSRT and Montgomery *combined* to increase variation in the training data. Without any further hyperparameter tuning, SCAN improves the IoU on two lungs to $95.1\% \pm 0.43\%$ on the JSRT evaluation set, and $93.0\% \pm 1.4\%$ on the Montgomery evaluation set, a significant improvement compared with when training on JSRT development set alone.

**Qualitative Comparison.** Fig. 6 shows the qualitative results from these two experiments. The failure cases in the middle row by our FCN reveals the difficulties arising from CXR images' varying contrast across samples. For example, the apex of the ribcage of the rightmost patient's is mistaken as an internal rib bone, resulting in the mask "bleeding out" to the black background, which has a similar intensity as the lung field. Vascular structures near mediastinum and anterior rib bones (which appears very faintly in the PA view CXR) within the lung field can also have similar intensity and texture as the exterior boundary, leading the FCN to make the drastic mistakes seen in the middle two columns. SCAN significantly improves all of the failure cases and produces much more realistic outlines of the organs. We also notice that adversarial training sharpens the segmentation of costophrenic angle (the sharp angle at the junction of ribcage and diaphragm). Costophrenic angles are important in diagnosing pleural effusion and lung hyperexpansion, among others.

Fig. 7 compares SCAN with the current state-of-the-art [16] qualitatively. We restrict the comparison to lung fields only as [16] only supports lung field segmentation. SCAN generates more accurate lung masks especially around costophrenic angles when tested on the same patient population (left two columns of Fig. 7). SCAN also generalizes better to a different population in the Montgomery dataset (right two columns of Fig. 7) whereas [16] struggles with domain shift.

|  | IoU (Both Lungs) |
|---|---|
| Ours (SCAN) | **91.4%** $\pm 0.6\%$ |
| Ours (FCN) | 87.1% $\pm 0.8\%$ |
| Registration [16] | 90.3% $\pm 0.5\%$ |

Table 3: Performance on the full Montgomery dataset using models trained on the full JSRT dataset. Compared with the JSRT dataset, the Montgomery dataset exhibits a much higher degree of lung abnormalities and varying imaging quality. This setting tests the robustness of the method in generalizing to a different population and imaging setting.

Our SCAN framework is efficient at test time, as it only needs to perform a forward pass through the segmentation network but not the critic network. Table 4 shows the run time of our method compared with [16] on a laptop with Intel Core i5. [16] takes much longer due to the need to search through lung models in the training data to find similar profiles, incurring linear cost in the size of training data. In clinical setting such as TB screening [3] a fast test time result is highly desirable.

|  | Test time |
|---|---|
| Ours (SCAN) | 0.84 seconds |
| Registration [16] | 26 seconds |
| Human | $\sim$2 minutes |

Table 4: Prediction time for each CXR image (resolution $400 \times 400$) from the Montgomery dataset on a laptop with Intel Core i5, along with the estimated human time.

## 4   Related Work and Discussion

**Lung Field Segmentation.** The current state-of-the-art method for lung field segmentation uses registration-based approach [16]. To build a lung model for a test patient, [16] finds patients in an existing database that are most similar to the test patient and perform linear deformation of their lung profiles based on key point matching.

**Semantic Segmentation with Convents.** Most semantic segmentation models use fully convolutional network (FCN) [9, 11, 12, 20]. We adapt FCNs to gray-scale CXR images under the stringent constraint of a very limited training dataset of 247 images. Our FCN departs from the usual VGG architecture and can be trained without transfer learning. Separately, U-net [18] and similar architectures are popular convolutional networks for biomedical segmentation and have been applied to neuronal structure [18] and histology images [21], but does not address global consistency directly as our method does. Our experiments show that they do not perform well for CXR segmentation.

**Computer-Aided CXR Diagnosis.** Accurate CXR segmentation can also directly lead to precise cardiothoracic ratio calculation [4]. There is a growing body of recent works that apply neural networks end-to-end on CXR images [22, 5, 23, 24]. These models directly output clinical targets such as disease labels. They often struggle with localizing to the correct lung regions [5] and do not offer interpretability to support their end-to-end prediction. Moreover, they generally require a large number of CXR images for training, which is not readily available or expensive to obtain for many clinical tasks. By leveraging multi-task learning, SCAN holds the promise to provide additional localization guidance to lung and heart regions in these end-to-end diagnosis models.

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

# Supplementary Material
# SCAN: Structure Correcting Adversarial Network for Organ Segmentation in Chest X-rays

## 1  Data Processing

We scale all images to $400 \times 400$ pixels, which retains sufficient visual details for vascular structures in the lung fields and the boundaries. Our experiments suggest that increasing the resolution to $800 \times 800$ pixels does not improve the segmentation performance, consistent with the observation in [2]. Due to the high variation in image contrast between dataset (Fig. (1) in the main manuscript), we perform instance normalization. Given an image $\boldsymbol{x}$ we normalize it with $\tilde{x}^{jk} := \frac{x^{jk} - \bar{x}}{\sqrt{\mathrm{var}(\boldsymbol{x})}}$, where $\bar{x}$ and $\mathrm{var}(\boldsymbol{x})$ are the mean and variance of pixels in $\boldsymbol{x}$, respectively. In this way we do not use statistics from the whole dataset. Data augmentation by rotating and zooming images, did not improve results in our preliminary experiments. We thus did not apply any data augmentation.

In post-processing, we fill in any hole in the predicted mask, and remove the small patches disjoint from the largest mask. We observe that in practice this is important for the prediction output of FCN, but does not affect the evaluation results of SCAN.

## 2  Further Training Protocols

We pre-train the segmentation network using only the pixel-wise loss $J_s$ (Eq. (1) in the main manuscript), which also gives faster training than the full adversarial training, as training the segmentation network using pixel losses involves forward and backward propagation through the segmentation network only but not the critic network. We use Adam optimizer with learning rate 0.0002 to train all models for 350 epochs, where an epoch is defined as a pass over the full training set. We use mini-batch size 10. When training involves critic network, we perform 5 optimization steps on the segmentation network for each optimization step on the critic network. Our training takes place on a machine equipped with a Titan X GPU.

## 3  Futher Related Work

Existing work on lung field segmentation broadly falls into three categories [12]: (1) Rule-based systems apply pre-defined set of thresholding and morphological operations that are derived from heuristics [5]. (2) Pixel classification methods classify the pixels as inside or outside of the lung fields based on pixel intensities [15, 6, 7, 1]. (3) More recent methods are based on deformable models such as Active Shape Model (ASM) and Active Appearance Model [4, 3, 10, 11, 9, 13, 8, 14]. Their performance can be highly variable due to the tuning parameters and whether shape model is initialized to the actual boundaries. Also, the high contrast between rib cage and lung fields can cause the model to be trapped in local minima. Our approach uses convolutional networks to perform end-to-end training from images to pixel masks without using ad hoc features. The proposed adversarial training further incorporates structural regularities in a unified framework.

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
