# OpenReview forum: "SCAN: Structure Correcting Adversarial Network for Organ Segmentation in Chest X-rays"
_MIDL.amsterdam/2018/Conference — Submitted to MIDL 2018_

### Review · AnonReviewer1 · 2018-05-04
**An important reference is missing**

**Rating:** 2
**Confidence:** 2

**Review:**

The paper proposes an automatic approach for Chest X-ray segmentation. The approach is based on Fully Convolutional Network which is trained with adversarial loss. The paper is well written and easy to follow.

My main concern with the paper is the way the main contribution is presented. From the introduction section, it seems like the paper proposes an architectural contribution: the use of adversarial loss in a FCN to improve segmentation results. However, very similar idea has been published in the context of Computer Vision in [1]. [1] is not mentioned in the current version of the paper, it should be properly referenced and discussed in the introduction section. What is novel in SCAN w.r.t [1]? It feels like it would be better to frame the paper as an application of [1] to X-rays rather than as an architectural contribution. Moreover, the ideas of [1] have been already applied in the context of medical imaging [2], [3] and [4].

Details on training, test and validation splits should be included in the paper. How many patients are in each split?

Questions to Tables 1 and 2:
1. Do the results represent cross validation results? If so, what is the value of k in k-fold cross validation?
2. How the dice and IoU are computed? Do the numbers represent mean of per subject Dice/IoU or per test set Dice/IoU?
3. Is the computation of the metric performed on downsampled annotations (400x400) or the output of network is upsampled to the original resolution before computing the metrics?

[1] https://arxiv.org/pdf/1611.08408.pdf
[2] https://arxiv.org/pdf/1706.09318.pdf
[3] https://arxiv.org/pdf/1707.03195.pdf
[4] https://arxiv.org/pdf/1707.08037.pdf

**Special Issue:**

No

---

### Review · AnonReviewer2 · 2018-05-06
**Is it necessary to use the same FCN model as the critic network?**

**Rating:** 2
**Confidence:** 2

**Review:**

This paper proposes to use a FCN-based adversarial network to segment lung and heart in Chest X-ray image. This paper is clear to read through except the related works should be ideally placed in or after introduction, and the experiment setups in the appendix is better to be placed in the paper.

First, I share AnonReviewer1's view that the technique presented in this paper is similar to existing methods in literature.

Second, from eq. (1), it is easy to see that the whole critic network serves as a regularization term to train the segmentation network.  Therefore, there is a trade-off between the complexity of the critic network versus the performance gain that can be explored.  For example, the gain from FCN to SCAN can be increased by how much if the critic network model size is increased from x to y in a few steps. It can also tell at which size the performance gain saturates.  I personally believe mirroring the same FCN model (~250K parameter) is overkill because the 4-channel predicted mask does not seem to be as informative as the original X-ray images.











**Special Issue:**

No

---

### Review · AnonReviewer3 · 2018-05-09
**The authors propose a structure correcting adversarial network (SCAN) for automated segmentation of the lung fields and the heart in CXR images. Specifically, a critic network is combined with FCN to learn high-order consistency between reference masks and FCN outputs. Experimental results show that train the critic network and FCN jointly can effectively improve the segmentation performance.**

**Rating:** 3
**Confidence:** 3

**Review:**

Pros:
The paper is well-written and easy to follow.

The method is technically sound and has a real impact in the clinical setting.

The experimental results support the conclusion that the proposed method could potentially reach the human-level performance.


Cons:
The integration of a critic network to improve the performance of a segmentation networks is not new, considering several approaches have been proposed in both fields of medical and natural image segmentation, e.g., SegAN [Xue et al., 2017] and [Luc et al., 2016]. The authors should clearly indicate their novelty compared with these existing methods.

It would be interesting to present also experimental results obtained by SCAN trained on Montgomery and tested on JSRT to evaluate if the proposed method could learn more from difficult cases.


**Special Issue:**

No

---

### Decision · Program_Chairs · 2018-05-15
**Paper117 Acceptance Decision**

Reject